# Impact of the COVID-19 pandemic on intra-household gender disparities in the Middle East and North Africa region: A scoping review protocol

Malak Ghezzawi[1], Sasha Fahme[1], Salpy Naalbandian[2], Jocelyn DeJong[1]*, WOMENA Study Group[1¶]

1 Faculty of Health Sciences, American University of Beirut, Beirut, Lebanon, 2 University Libraries, American University of Beirut, Beirut, Lebanon

¶ Membership of the WOMENA Study Group is provided in the Acknowledgments.
* Jd16@aub.edu.lb

**Data Availability Statement:** No datasets were generated or analyzed during the current study. All relevant data are available from within the

## Abstract

### Introduction

The Middle East and North Africa (MENA) is a global outlier both for its low female labor force participation and investment in early childhood development services, and consistently ranks lowest on global rankings of gender parity. While the impact of the COVID-19 pandemic on global gender inequity has been acknowledged, specific challenges faced by partnered-women in MENA are underexplored. Additionally, with over half of the region affected by conflict and displacement, exploring these impacts sheds light on understudied gender disparities in humanitarian contexts during the pandemic. This scoping review aims to examine intra-household gender disparities during the COVID-19 pandemic, adding to a more comprehensive understanding of this particular public health emergency's worldwide ramifications. The hypothesis is that the pandemic has exacerbated women's pre-existing constraints in the MENA region, worsening gender inequities in economic empowerment, healthcare access, and general well-being.

### Methods

This review will explore intra-household gender disparities in decision-making, household labor division, economic independence, health, and access to resources and services during the COVID-19 pandemic that have been reported to have been affected by COVID-19 globally. Following PRISMA guidelines for Scoping Reviews, a comprehensive search will be conducted in APA PsycINFO, Arab World Research Source: Al Masdar, EconLit, Global Health, MEDLINE, Scopus, and Web of Science Core Collection, in consultation with an information specialist. Studies in English, French and Arabic from January 2020 to August 2024 will be included. Four independent reviewers will screen studies, and data will be charted, coded, and narratively synthesized.

manuscript as well as a supplemental information file.

**Funding:** This study is supported by a 2-year grant from the International Development Research Centre (IDRC) in Canada under the reference number 110025. The funders had no role in study design, data collection and analysis, decision to publish, or preparation of the manuscript.

**Competing interests:** The authors have declared that no competing interests exist.

**Abbreviations:** MENA, Middle East and North Africa; LMICs, low- and middle- income countries; WEI, Women Gender Equality Index; GGPI, Gender Gap Performance Index; PRISMA-ScR, PRISMA guidelines for Scoping Reviews; PRISMA-P, PRISMA guidelines for protocols; WHO, World Health Organization; GBV, gender-based violence.

## Discussion

This review is expected to shed light on gender disparities in healthcare accessibility, mobility, and decision-making during the COVID-19 pandemic across low- and middle-income countries in the MENA region, adding to the global literature.

## Introduction

The COVID-19 pandemic has significantly reshaped various aspects of society, with notable consequences on gender disparities within households [1]. Its impact encompasses employment losses, an increase in unpaid labor, educational disruptions, food security challenges, unmet health needs, and personal safety concerns [2–5] all of which affect gender relations within families and households. Households are the epicenter of social interactions, work, and caregiving responsibilities, the latter of which may disproportionately burden women [6]. Global evidence highlights women's heightened vulnerability to social and economic hardships during the COVID-19 pandemic [7–9]. Understanding the pandemic's impact on gender dynamics within households is crucial for identifying challenges and opportunities to propose targeted interventions to safeguard women's health, empowerment, and well-being for a more resilient and equitable society [6].

In the Middle East and North Africa (MENA) region, the COVID-19 pandemic has been reported to have exacerbated pre-existing gender disparities, but until now these effects have been relatively understudied [10, 11]. The region is a global outlier both in terms of having the lowest female labor force participation [12, 13], and for its low investment in early childhood development [14], which is critical to support women's employment [10, 11]. The region ranks fourth among the five regions covered by the SDG Gender Index with an average score of 60.8 [15]. Furthermore, MENA has the lowest score among the eight global regions on the Global Gender Gap Index, with a score of 62.6%, significantly trailing the highest-ranking region, Europe, which scores 76.3% [16]. In terms of economic participation and opportunity, MENA's score of 44% is the second-lowest after Southern Asia's 37.2%, compared to other regions that score above 65%, with North America leading at 77.6% [16]. Specific country rankings within MENA reveal stark disparities: Jordan ranks 5th regionally but 126th globally, Tunisia 6th regionally and 128th globally, and Lebanon and Egypt 8th and 10th regionally, and 132nd and 134th globally, respectively [16]. The UN Women's Gender Gap Performance Index (GGPI) and Women's Empowerment Index (WEI) further underscore significant disparities [17]. Countries like Egypt, Iraq, Jordan, Lebanon, Tunisia, and Yemen rank low on both indices, highlighting widespread gender inequality in economic participation, education, health, and political empowerment [17]. Meanwhile, other low- and middle- income countries (LMICs) in MENA, including Algeria, Djibouti, Libya, Morocco, Palestine, Somalia, Sudan, and Syria, have incomplete data, reflecting gaps in information and complicating efforts to address gender disparities [17]. This pattern underscores the persistent challenges faced by women in the MENA region and emphasizes the need for targeted interventions to improve gender equality [16, 17].

By September 2021, gender disparities in employment that predated the pandemic became more pronounced both globally and within the MENA region [10, 11]. Informal caregiving responsibilities increased during the pandemic because of school and childcare closures [18] and lockdowns enforced to slow the spread of the pandemic. Even among remote-working

couples, an increase in gendered domestic work has been reported both globally and in MENA, exacerbating pre-existing differences [18, 19].

Available evidence suggests that amidst lockdowns, economic instability, and stress, couples experienced elevated anxiety both globally and in MENA [20–23]. Insufficient support structures and increased financial dependence on men exacerbated these stressors among women [20–23]. Gender gaps in income loss may have contributed to a surge in intimate partner violence (IPV) both worldwide [24] and in MENA [25]. Focusing on LMICs in MENA, this scoping review aims to uncover how the pandemic has impacted intra-household gender disparities, contributing to a comprehensive understanding of the global implications of the public health emergency. Our findings may guide targeted interventions and public policies to protect women during future crises.

The aim of this scoping review is to map and narratively explore the available literature on the association between the COVID-19 pandemic and intra-household gender inequality among LMICs in the MENA region. This scoping review asks the research question: "What are the types, characteristics, and implications of intra-household gender disparities identified during the COVID-19 pandemic in LMICs in the MENA region?" This review aims to characterize women's experiences and unmet needs with respect to economic independence, health, and unpaid labor/care during the pandemic.

## Methods

A scoping review will be undertaken to compile hits from seven databases and including peer-reviewed articles in Arabic, English, and French, which are the three most spoken languages in the region. These databases encompass both global and region-specific literature. MEDLINE (OVID) and Global Health (CAB) provide extensive coverage of medical and health-related research globally. APA PsycINFO (EBSCO) is essential for understanding changes in mental health, family dynamics, and gender roles during the pandemic. EconLit with Full Text (EBSCO) captures the economic dimensions of intra-household dynamics and gender inequality. Arab World Research Source: Al-Masdar (EBSCO) focuses on literature specific to the Arab region, offering insights that complement global databases. Scopus and Web of Science Core Collection ensure a broad and inclusive range of peer-reviewed literature across multiple disciplines, including social sciences, health, and humanities, covering both global and regional studies.

The development of this scoping review protocol was guided by the Preferred Reporting Items for Systematic Reviews and Meta-Analyses extension for Scoping Reviews (PRISMA-ScR) [26], PRISMA guidelines for protocols (PRISMA-P) [27], the updated methodological guidance for the conduct of scoping reviews [28], and in consultation with an information specialist with expertise in library science.

### Inclusion criteria

This scoping review focuses on peer-reviewed, data-driven articles that address the impact of the COVID-19 pandemic on intra-household gender disparities in LMICs within the MENA region, as defined by the World Bank (Algeria, Djibouti, Egypt, Iraq, Jordan, Lebanon, Libya, Morocco, Occupied Palestinian territory, Somalia, Sudan, Syria, Tunisia, and Yemen) [29]. Several of these countries, including Egypt, Iraq, Jordan, Lebanon, Libya, Morocco, Somalia, Sudan, Syria, and Yemen, are experiencing varying degrees of conflict and instability. These conflicts may exacerbate gender disparities by changing gender roles and influencing socioeconomic tensions within households [30–32]. Conflict-affected settings are underrepresented in the COVID-19 literature, and this review will shed light on gender disparities arising in the

context of compounding public health and humanitarian crises. The review will include both quantitative and qualitative studies, encompassing primary studies and secondary analyses that appropriately cite primary data sources, ensuring they are empirical and contribute directly to understanding gender disparities within households during the pandemic. Articles published in English, French, or Arabic and covering the period from January 2020 to August 2024 will be included. Themes and outcomes for the included studies must address at least one of the following defined areas:

1. **Gender roles**: The roles and responsibilities assigned to men and women within households during the pandemic, including characteristics of caregiving and employment roles. This theme is essential because the pandemic has significantly disrupted established gender roles, impacting family structures and gender equity. Understanding these reported roles and responsibilities is crucial for identifying shifts in household dynamics and evaluating their potential long-term effects on gender norms [24, 33].

2. **Decision-making power**: The authority within household decision-making during the pandemic, including shifts in influence over financial, health, and educational decisions. This theme is important for characterizing changes in gender relations and autonomy within households, highlighting how the pandemic has affected gender equity and individual roles in household management [24, 34].

3. **Division of household labor**: The allocation of household tasks during the pandemic, such as cleaning, cooking, and child-rearing, and their impact on gendered task distribution. Examining this theme helps to highlight any increases in the burden on women, who typically bear a larger share of domestic responsibilities. It is crucial for assessing how the pandemic has altered gendered divisions of labor and its implications for gender equality [24, 34].

4. **Economic dependence**: The degree of women's economic reliance on men during the pandemic, including changes in income sources and employment status, and their impact on financial dependency. The economic impact of the pandemic has been profound, with many women experiencing job losses or reduced income. This theme seeks to evaluate financial vulnerabilities and dependencies exacerbated by the pandemic, particularly regarding gender disparities in financial independence [33].

5. **Gender based violence**: Incidence of gender-based violence within households during the pandemic, examining contributing factors and consequences. This theme is critical for understanding the implications of the pandemic on safety and gendered power dynamics, as studies reporting changes in violence prevalence will illuminate the effects of increased stress and economic hardship [10].

6. **Health**: Physical and mental health outcomes within households during the pandemic, including management of health concerns and gender disparities in health outcomes. This theme aims to investigate how the pandemic has differentially affected men and women's health, emphasizing studies that report changes in health outcomes as a result of the crisis [35].

7. **Access to resources and services**: The availability and utilization of essential resources and services, such as healthcare and education, and their effects on gender roles and relationships within households during the pandemic. This theme seeks to explore disruptions in access and how these changes have influenced intra-household gender dynamics and disparities [10].

## Exclusion criteria

Excluded texts will be those conducted in countries which do not fulfill our inclusion criteria. Grey literature, systematic reviews, meta-analyses, letters, books, book chapters, and conference abstracts will be excluded. Articles that report outcomes different than our outcomes of interest will be excluded.

## Information sources and search strategy

The search will be conducted covering the period from January 2020 to August 2024. This is to capture studies conducted during the pandemic but published after the WHO-specified pandemic end date. Seven databases (MEDLINE (OVID), Arab World Research Source: Al-Masdar (EBSCO), APA PsycINFO (EBSCO), EconLit with Full Text (EBSCO), Global Health (CAB), Scopus, Web of Science Core Collection) will be used for this review. Refer to **S1 Appendix** for the MEDLINE search strategy. The search strategies will be developed in collaboration with an information specialist to ensure thorough and effective retrieval of relevant literature.

## Data management

Search results will be imported into Covidence software [36] and de-duplicated. All articles will be screened in Covidence.

## Selection process

Before commencing the actual selection process by the entire team, a pilot testing phase will be conducted to ensure consistency and clarity in applying the eligibility criteria. The pilot testing will follow the framework described below: a random sample of 25 titles and abstracts will be selected from the identified records using a random number generator. Two reviewers will independently screen the random sample of titles and abstracts using the pre-defined eligibility criteria. The team will meet to discuss any discrepancies or uncertainties that arise during the pilot testing. Based on the discussions and feedback, modifications to the eligibility criteria and definitions/elaboration document will be made to ensure clarity and consistency. The team will only begin the full source selection process when a consensus agreement of 95% or greater is achieved among the reviewers during the pilot testing. This threshold ensures that the team is aligned in their understanding and application of the inclusion criteria.

The first stage of screening will entail four independent reviewers assessing both the title and abstract for peer-reviewed articles, in duplicates. Following the first screen, articles marked for full text review will be read by four independent reviewers to ensure alignment with the inclusion criteria. This will also be done in duplicates. In cases where there are disagreements between the reviewers regarding the inclusion of a particular study, the reviewers will attempt to reach a consensus through discussion and mutual agreement. If a consensus cannot be reached, a third reviewer (JD or SAF) will be involved to make the final decision on the inclusion or exclusion of the study. For excluded sources, the reasons for exclusion will be clearly stated, ensuring transparency and reproducibility in the review process.

## Data collection process and data extraction

A data extraction template will be used to compile information on categories of conceptualizations that appear in included papers and the bibliographic information of author(s), publication year, and publication title. Refer to **S2 Appendix** for the data extraction template. The development of the data extraction template will be led by one author and piloted using an

initial set of included papers (n = 25). Before starting the extraction process, the review team will ensure that 95% or greater agreement is achieved during the pilot testing phase. The data extraction will be done by four members of the review team. Reviewers will be open to extracting any relevant data that aligns with the scoping review questions, even if not initially included in the charting table.

## Data analysis

Data analysis will primarily involve descriptive mapping of the extracted results from the included sources. The focus will be on providing a clear overview of the key concepts, populations, characteristics, and outcomes related to intra-household dynamics during the COVID-19 pandemic. The analysis will involve mapping the occurrence of specific gender disparities such as those impacting gender roles, decision-making, and division of household tasks in each of the LMICs within the specified timeframe. Principles of framework synthesis will also be useful in organizing and categorizing findings/data from the included sources. By charting and sorting the extracted data against an a priori identified framework, the scoping review results will provide a structured overview of how the COVID-19 pandemic impacted gender inequality.

## Ethical considerations

As this review involves a synthesis and presentation of available resources, it does not require ethics approval. Results will be published in a peer-reviewed journal, developed into easily disseminated infographics and shared at international conferences.

## Status and timeline of the study

The search strategy for MEDLINE has been developed with the information specialist. The search on MEDLINE will be run along with that of other databases, followed by data screening (title-abstract and full-text) and data extraction within 3 months. Then, data analysis will be conducted, followed by the manuscript write-up.

## Discussion

Data on the impact of the COVID-19 pandemic on gender disparities in the MENA region are lacking. Global studies on intra-household dynamics and gender disparities during the COVID-19 pandemic have often excluded MENA countries [10, 37] and/or are focused on high-income settings [38], leading to a lack of empirical evidence from LMICs within this area. Where data exist, scoping reviews have focused on either broader populations beyond the MENA region [39], or specific aspects of intra-household dynamics, such as domestic violence [40]. For instance, Moyano et al.'s scoping review investigated gender considerations in global health and social protection policies during the pandemic, highlighting adverse effects on gender roles and increased inequalities, including gender-based violence (GBV), but did not address the MENA region or its specific challenges [39]. A systematic review on intimate partner violence in the Gulf Cooperation Council Arab nations found varying prevalence rates of GBV (9.0–45.5% physical abuse, 22–69% psychological abuse and 6.9–19.2% sexual abuse), but did not isolate the COVID-19 period [41]. Another systematic review on economic conditions, health outcomes, and GBV globally revealed challenges faced by employed women in LMICs worldwide, including high job losses, GBV, limited access to health services, and increased mental health issues such as anxiety [35]. However, most LMICs in MENA were overlooked by these systematic reviews [35, 41].

The MENA region stands out due to its particularly high levels of gender inequality, which predate the pandemic [10, 11]. As noted, countries in this region frequently rank poorly on global gender equality indices, reflecting deep-seated disparities in economic participation, educational attainment, and health [16, 17]. Additionally, ongoing conflicts in several MENA countries, such as Syria, Yemen, and Libya, may further compound the impact of the pandemic and intensify the challenges faced by women. Despite the prevalence of these issues, the region remains underrepresented in global studies, which tend to focus on high-income and/or non-humanitarian settings.

This scoping review seeks to address this imbalance by providing a focused examination of intra-household gender disparities across all LMICs within the MENA region. By concentrating on this region, the review aims to fill a significant gap in the global literature and offer insights into how cultural, socio-economic, and conflict-related factors uniquely shape gender disparities during the pandemic. The findings will contribute to a more nuanced understanding of the pandemic's impact on gender inequality and inform targeted interventions and policies.

## Limitations

The reliance on specific databases for literature search may result in the omission of relevant peer-reviewed articles that are not indexed within these databases. Cross-referencing the included papers will be conducted to mitigate this limitation. Moreover, our study focuses exclusively on peer-reviewed literature, thereby excluding potentially valuable insights from grey literature sources such as conference proceedings, dissertations, and reports. This exclusion might lead to a bias towards published works or omission of relevant studies that are not widely disseminated. Though limiting our inclusion criteria to longitudinal studies would be ideal to examine change, there is a paucity of cohort studies in the region both generally and with respect to the COVID-19 pandemic [42–44]. Thus, this review will include cross-sectional data on perceived changes as reported by study participants as well as longitudinal studies that might be available.

## Dissemination plans

The results will be published in a peer-reviewed publication and presented at international conferences. The team is continuing to engage with local and regional public sector actors throughout the review process, so the outcomes will be shared with them as well.

## Supporting information

**S1 Checklist. PRISMA-P (Preferred Reporting Items for Systematic review and Meta-Analysis Protocols) 2015 checklist: Recommended items to address in a systematic review protocol\*.**
(PDF)

**S1 Appendix. Search strategy.**
(DOCX)

**S2 Appendix. Data extraction Instrument.**
(DOCX)

## Acknowledgments

We would like to thank the WOMENA Study Group for supporting the study work (Stephen McCall, Hala Ghattas, Rita itani, Myriam Dagher, Ali Abboud, Nisreen Salti, Ghada Saad).

## Author Contributions

**Conceptualization:** Sasha Fahme, Jocelyn DeJong.

**Methodology:** Malak Ghezzawi, Salpy Naalbandian, Jocelyn DeJong.

**Supervision:** Sasha Fahme, Jocelyn DeJong.

**Visualization:** Malak Ghezzawi.

**Writing – original draft:** Malak Ghezzawi.

**Writing – review & editing:** Malak Ghezzawi, Sasha Fahme, Salpy Naalbandian, Jocelyn DeJong.

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
