## [Decision Letter · Decision Letter 0]

10 Jul 2024

PONE-D-24-14476Impact of the COVID-19 Pandemic on Intra-household Dynamics and Gender Inequality in the Middle East and North Africa region: A Scoping Review ProtocolPLOS ONE

Dear Dr. DeJong,

Thank you for submitting your manuscript to PLOS ONE. After careful consideration, we feel that it has merit but does not fully meet PLOS ONE’s publication criteria as it currently stands. Therefore, we invite you to submit a revised version of the manuscript that addresses the points raised during the review process.

We look forward to receiving your revised manuscript.

Kind regards,

Amin Nakhostin-Ansari

Academic Editor

PLOS ONE

“This study is supported by a 2-year grant from the International Development Research Centre (IDRC) in Canada under the reference number 110025.”

Reviewers' comments:

Reviewer's Responses to Questions

**Comments to the Author**

1. Does the manuscript provide a valid rationale for the proposed study, with clearly identified and justified research questions?

Reviewer #1: Yes

Reviewer #2: Yes

2. Is the protocol technically sound and planned in a manner that will lead to a meaningful outcome and allow testing the stated hypotheses?

Reviewer #1: Yes

Reviewer #2: Partly

3. Is the methodology feasible and described in sufficient detail to allow the work to be replicable?

Reviewer #1: Yes

Reviewer #2: Yes

4. Have the authors described where all data underlying the findings will be made available when the study is complete?

Reviewer #1: No

Reviewer #2: Yes

5. Is the manuscript presented in an intelligible fashion and written in standard English?

Reviewer #1: Yes

Reviewer #2: Yes

6. Review Comments to the Author

You may also provide optional suggestions and comments to authors that they might find helpful in planning their study.

Reviewer #1: This is a review of a scoping study protocal on the gendered effects of the pandemic in 11 LMIC countries in the MENA-region. I think the study potentially could offer important insights but further attention is needed to several issues before proceeding with the scoping review.

Framing/introduction:

1. The authors start quite broad in relation to the pandemic, which in my view could move straight away to the gendered aspects of the pandemic. Primarily because I am missing a logic/justifciation for the need to focus on the MENA region in the Introduction.

2. Moreover, it is not entirely clear from the Introduction which aspects of gender inequality will be considered. Several areas of concern are noted, including women’s (reproductive) health, but the authors then state they will focus on intra-household gender dynamics. In such a short overview, it would be helpful to have an argument that clearly outlines the issues at stake specific to intra-household gender dynamics and the need for further study.

Case study selection/inclusion criteria:

3. The authors focus on 11 LMIC countries in the MENA region. How many countries are there in the MENA region? And why focus on these 11? In other words, greater justification for case study selection (inclusion criteria) is needed.

4. (Greater) justification of the outcome variables of interest is needed. Why these variables and not others? What does this mean for which studies may be included/excluded in the scoping review?

5. The inclusion of studies in English, French and Arabic is a strong advantage of the study and should be emphasized in the Abstract.

6. The authors argue that searching until 3 months past the end of the pandemic allows for studies on the pandemic published after its end to still be included. Arguably, many studies are still coming out, particularly those with longitudinal and/or comparative data. Would it not be better to have the longest time frame possible, including the entire year of 2023 and the first months of 2024? The justification therefore does not currently align with the timeframe.

7. Greater explanation is needed for how many databases were considered and why the choice was given to focus on these 7.

8. Greater detail is needed on data management – how will articles be assessed/screened?

Discussion:

9. To clarify the contribution to the literature, I would encourage the authors to do more to explain why greater insights on the MENA region are needed. While in principle I agree that there is a significant gap (and an over-abundance of empirical data on North America and Europe, the existence of a gap is in and of itself insufficient justification for a study.

Reviewer #2: This manuscript describes an interesting and worthwhile scoping review. I think it is promising but have a few small concerns.

1. Two small textual issues: if MENA is spelled out then LMIC also should be spelled out at its first appearance, and “worldwide” at the end of the first sentence of the second last paragraph of the Introduction should be removed.

2. It's unclear why the literature search should end at July 2023. The issue here is not when the pandemic ended but when research on it was being published. Surely relevant work carried out in 2022 and early 2023 could still going through the data analysis, write-up, and editorial review processes long after the pandemic ended. I would be inclined to extend the study period another year to reduce the risk of excluding relevant articles. If relevant articles aren't being published after July 2023 the researchers will be able to determine this empirically.

3. Not much detail is given on how the various themes (e.g.., "changes in gender roles") will be defined at the inclusion stage. Some of them are a little vague (e.g., "health") so to ensure replicability it would be good to be clearer on defining each theme.

4. The research question is about change in household dynamics but there seems to be no requirement that the studies to be reviewed are longitudinal or panel studies. Without data of this kind it's unclear how change can be assessed, except unreliably by recollection or by comparison to some external baseline data. The manuscript should specify whether (A) all included studies will be longitudinal or panel; and (B) explain why other kinds of data should be included to evaluate "change" if they are to be included; and (C) explain how findings of "change" based on different kinds of data will be integrated in the review.

7. PLOS authors have the option to publish the peer review history of their article (what does this mean?). If published, this will include your full peer review and any attached files.

Reviewer #1: No

Reviewer #2: No

---

## [Author Response · Author response to Decision Letter 0]

27 Aug 2024

Journal: PLOS One

PONE-D-24-14476

 24-08-2024

Dear Editor, 

On behalf of all co-authors, we would like to thank you and the reviewers for the time spent on reviewing our work, as well as the valuable comments that helped us improve the quality of our manuscript titled “Impact of the COVID-19 Pandemic on Intra-household Dynamics and Gender Inequality in the Middle East and North Africa region: A Scoping Review Protocol”. All comments have been addressed, as can be seen in the point-by-point response to the comments by reviewers below.

All authors confirm that the manuscript is not under consideration by any other journal and that authors have no conflict of interest. All authors have read and approved the submission of this revised manuscript.

Yours sincerely, 

Jocelyn DeJong, PhD

Associate Provost and Professor at the Faculty of Health Sciences, American University of Beirut, Beirut, Lebanon 

Email: Jd16@aub.edu.lb

Journal requirements

Thank you for your comment. We have carefully reviewed the PLOS ONE style requirements and ensured that our revised protocol complies with these guidelines.

“This study is supported by a 2-year grant from the International Development Research Centre (IDRC) in Canada under the reference number 110025.”

Thank you for your comment. The following statement ‘The funders had no role in study design, data collection and analysis, decision to publish, or preparation of the manuscript’ has been added to the manuscript.

Thank you for your comment. Since our manuscript is a scoping review protocol, it does not involve original data collection; there are no raw datasets to be shared. We understand and comply with your open data policy where applicable. If there are specific requirements or additional steps needed for making the review protocol and its components accessible, please let us know, and we will ensure that these requirements are met. 

Reviewer #1

We would like to thank the Reviewer for the time reviewing our work and the valuable comments.

This is a review of a scoping study protocol on the gendered effects of the pandemic in 11 LMIC countries in the MENA-region. I think the study potentially could offer important insights but further attention is needed to several issues before proceeding with the scoping review.

Framing/introduction:

1. The authors start quite broad in relation to the pandemic, which in my view could move straight away to the gendered aspects of the pandemic. Primarily because I am missing a logic/justification for the need to focus on the MENA region in the Introduction.

Thank you for this comment. We have revised the Introduction to more directly address the gendered aspects of the pandemic (lines 80-100).

2. Moreover, it is not entirely clear from the Introduction which aspects of gender inequality will be considered. Several areas of concern are noted, including women’s (reproductive) health, but the authors then state they will focus on intra-household gender dynamics. In such a short overview, it would be helpful to have an argument that clearly outlines the issues at stake specific to intra-household gender dynamics and the need for further study.

Thank you for this insightful comment. We’ve revised the manuscript to indicate that we will be focusing on gender disparities in several areas of women’s health, wellbeing, and economic in/dependence, including division of household labor, decision-making power, and access to resources and services (lines 69-78; lines 109-122).

Case study selection/inclusion criteria:

3. The authors focus on 11 LMIC countries in the MENA region. How many countries are there in the MENA region? And why focus on these 11? In other words, greater justification for case study selection (inclusion criteria) is needed.

This study is part of a larger study which focuses on the 11 countries chosen. We accept the reviewers’ comments about the justification of these 11 countries in relation to this study in particular, and have decided to include the other three low-and-middle-income countries in the region that had previously been excluded. The focus of this scoping review has been expanded to now include all 14 low- and middle-income countries (LMICs) as defined by the World Bank in the MENA region: Algeria, Djibouti, Egypt, Iraq, Jordan, Lebanon, Libya, Morocco, Palestine, Somalia, Syria, Sudan, Tunisia, and Yemen. This broader inclusion ensures that the review comprehensively represents the economic and geographic diversity within this category of countries in the region. The manuscript has been edited accordingly (lines 143-149).

4. (Greater) justification of the outcome variables of interest is needed. Why these variables and not others? What does this mean for which studies may be included/excluded in the scoping review?

Thank you for your valuable feedback. We appreciate the opportunity to provide further justification for the outcome variables of interest and their relevance to our scoping review. The outcome variables chosen for this scoping review are carefully selected to capture a comprehensive understanding of how the COVID-19 pandemic has impacted intra-household gender disparities in LMICs within the MENA region. These variables were identified based on their significance in the existing literature and their relevance to understanding gender inequality within households during the pandemic.

• Gender Roles: This variable is essential because the pandemic has caused significant disruptions to traditional gender roles and responsibilities within households. Understanding these reported changes is crucial for identifying ways in which to protect women’s well-being during public health shocks. 

• Decision-Making Power: Reported shifts in decision-making authority within households can indicate changes in power dynamics. These shifts are critical for understanding how the pandemic has affected gender relations and individual autonomy within households.

• Division of Household Labor: The allocation of household tasks and unpaid care is a key indicator of gender equity. Investigating reported changes in this area helps highlight any increases in the burden on women, who often handle a larger share of domestic work.

• Economic Dependence: The pandemic’s economic impact has been profound, with many women losing their jobs or experiencing reduced income. Examining these reported changes helps to understand the financial vulnerabilities and dependencies that have been exacerbated by the pandemic.

• Health: Both physical and mental health impacts are crucial for understanding the overall well-being of individuals within households and gender differences. The pandemic has placed significant stress on health systems and health status, making this an important area of focus.

• Access to Resources and Services: Access to essential resources such as healthcare, education, and social support services has been disrupted during the pandemic. Understanding these reported disruptions is vital for assessing the broader impact on intra-household gender parity.

These outcome variables were selected to ensure a comprehensive analysis of the multifaceted impacts of the COVID-19 pandemic on intra-household gender equity. By focusing on these key areas, our scoping review aims to capture a wide range of relevant studies that address these critical aspects of gender inequality. This approach allows us to include studies that provide empirical data on these variables, thereby contributing to a more holistic understanding of the pandemic's impact on gender dynamics within households.

Further detail on these outcome variables and the justification for their inclusion has now been added to the text (lines 158-204). 

5. The inclusion of studies in English, French and Arabic is a strong advantage of the study and should be emphasized in the Abstract.

Thank you for your comment and we agree that this is a strength. This point has been added in the abstract (lines 58-59).

6. The authors argue that searching until 3 months past the end of the pandemic allows for studies on the pandemic published after its end to still be included. Arguably, many studies are still coming out, particularly those with longitudinal and/or comparative data. Would it not be better to have the longest time frame possible, including the entire year of 2023 and the first months of 2024? The justification therefore does not currently align with the timeframe.

Thank you for your valuable feedback. We appreciate your observation regarding the timeframe for including studies related to the pandemic. We had chosen this cut-off date, three months after the WHO declared the COVID-19 pandemic to be over, to capture studies conducted and submitted for publication during the pandemic period. In response to your comment, we agree that extending the search period would provide a more comprehensive inclusion of relevant studies published after that date, particularly those with longitudinal and comparative data. We will therefore revise our search strategy to include studies published up until August 2024. This extension will ensure that our review captures the most recent and relevant research related to the pandemic and its impacts. For practical reasons due to research funding, we are not able to extend the period beyond that. The manuscript has been edited accordingly (line 155).

7. Greater explanation is needed for how many databases were considered and why the choice was given to focus on these 7.

Thank you for your comment. For this review, based on the guidance of our information scientist, seven databases will be used to ensure a thorough search encompassing health, psychological, social, economic, and regional perspectives:

MEDLINE (OVID) and Global Health (CAB) offer extensive coverage of medical and health related research.

APA PsycINFO (EBSCO) covers literature in psychology and related disciplines, which is essential for understanding changes in mental health, family dynamics, and gender roles during the pandemic.

EconLit with Full Text (EBSCO) is the primary database for economic literature. Its inclusion ensures a more comprehensive understanding of the economic dimensions of intra-household dynamics and gender inequality.

Arab World Research Source: Al-Masdar (EBSCO) focuses on literature specific to the Arab region, offering insights that might not be covered in global databases.

Scopus and Web of Science Core Collection offer comprehensive coverage of the peer-reviewed literature across multiple disciplines, including social sciences, health, and humanities, ensuring a broad and inclusive range of studies.

The manuscript has been edited accordingly with more explanation of these chosen databases (lines 128-136).

8. Greater detail is needed on data management – how will articles be assessed/screened?

Thank you for your comment. Details on how the articles will be assessed and screened are present in the 'Selection Process' and 'Data Collection Process and Data Extraction' paragraphs that follow the 'Data Management' paragraph (lines 219-257).

Discussion:

9. To clarify the contribution to the literature, I would encourage the authors to do more to explain why greater insights on the MENA region are needed. While in principle I agree that there is a significant gap (and an over-abundance of empirical data on North America and Europe, the existence of a gap is in and of itself insufficient justification for a study.

Thank you for your valuable feedback. We have addressed your concerns in the Discussion section. We emphasize that the MENA region is characterized by significant gender inequality, as highlighted by global gender parity rankings such as the SDG Gender Index, Global Gender Gap Index, UN Women’s Gender Gap Performance Index (GGPI), and Women's Empowerment Index (WEI). The COVID-19 pandemic has exacerbated these inequalities, although to our knowledge, a systematic examination of the literature has not been conducted. Furthermore, global studies on gendered aspects of the pandemic often overlook the MENA region, leading to an imbalance in empirical evidence, especially from LMICs. Examples of global studies that exclude this region are cited in the text. In addition, over half of the LMICs included in this scoping review are experiencing protracted conflict and/or forced displacement. Examining gender disparities in the context of these compounding crises and underrepresented humanitarian settings is crucial for improving women’s health and well-being in unstable settings. These points have been incorporated into the discussion to strengthen the rationale for focusing on the MENA region (Lines 303-309).

Reviewer #2

We would like to thank the Reviewer for the time spent on our work and the valuable comments.

1. Two small textual issues: if MENA is spelled out then LMIC also should be spelled out at its first appearance, and “worldwide” at the end of the first sentence of the second last paragraph of the Introduction should be removed.

Thank you for your comment. These points have been addressed in the manuscript.

2. It's unclear why the literature search should end at July 2023. The issue here is not when the pandemic ended but when research on it was being published. Surely relevant work carried out in 2022 and early 2023 could still going through the data analysis, write-up, and editorial review processes long after the pandemic ended. I would be inclined to extend the study period another year to reduce the risk of excluding relevant articles. If relevant articles aren't being published after July 2023 the researchers will be able to determine this empirically.

Thank you for your valuable feedback. We appreciate your observation regarding the timeframe for including studies related to the pandemic. We had chosen this cut-off date because it was 3 months after the official end of the pandemic, as declared by WHO. In response to your comment, we agree that extending the search period would provide a more comprehensive inclusion of relevant studies published after that date, particularly those with longitudinal and comparative data that may have been published after the pandemic's official end. We will therefore revise our search strategy to include studies published up until August 2024. This extension will ensure that our review captures the most recent and relevant research rel

---

## [Decision Letter · Decision Letter 1]

11 Sep 2024

PONE-D-24-14476R1Impact of the COVID-19 Pandemic on Intra-household Gender Disparities in the Middle East and North Africa region: A Scoping Review Protocol

PLOS ONE

Dear Dr. DeJong,

Thank you for submitting your manuscript to PLOS ONE. After careful consideration, we feel that it has merit but does not fully meet PLOS ONE’s publication criteria as it currently stands. Therefore, we invite you to submit a revised version of the manuscript that addresses the points raised during the review process.

*Comment from the PLOS editorial office: When you resubmit your manuscript*, *could you please ensure that you remove "(Please note that this is a revised title from the initial submission)" from the title of your submission?*

We look forward to receiving your revised manuscript.

Kind regards,

Amin Nakhostin-Ansari

Academic Editor

PLOS ONE

Journal Requirements:

Reviewers' comments:

Reviewer's Responses to Questions

**Comments to the Author**

1. Does the manuscript provide a valid rationale for the proposed study, with clearly identified and justified research questions?

Reviewer #1: Yes

Reviewer #2: Yes

2. Is the protocol technically sound and planned in a manner that will lead to a meaningful outcome and allow testing the stated hypotheses?

Reviewer #1: Yes

Reviewer #2: Yes

3. Is the methodology feasible and described in sufficient detail to allow the work to be replicable?

Reviewer #1: Yes

Reviewer #2: Yes

4. Have the authors described where all data underlying the findings will be made available when the study is complete?

Reviewer #1: Yes

Reviewer #2: Yes

5. Is the manuscript presented in an intelligible fashion and written in standard English?

Reviewer #1: Yes

Reviewer #2: Yes

6. Review Comments to the Author

You may also provide optional suggestions and comments to authors that they might find helpful in planning their study.

Reviewer #1: This is a review of a revised scoping study protocol. I think the authors have given careful consideration to the comments from myself and another reviewer, which have significantly improved the intended scoping review. I think this has the potential to be a very important study, providing a much needed contribution to the literature. I would be happy for the authors to proceed, with just two minor comments.

As noted by R2, the first mention of LMIC should be spelled out fully. This currently takes place on p4, but is not spelled out until p5.

On p7, the authors provide a careful outline of the themes to be included in the review. However, no references are provided. These should be included to understand the basis for including these themes.

Reviewer #2: The revised versions is largely responsive to my original comments. I'll discuss each in turn, numbered in the same way.

1. Dealt with appropriately.

2. This is a positive change and should increase the yield of useable studies.

3. These are much improved. I was only seeking clarification on what each theme was intended to cover, and the new text also provides a rationale for doing so, but I see the other reviewer requested this so I am satisfied with it.

4. I appreciate that there may be a paucity of longitudinal studies and that it is therefore reasonable to include cross-sectional studies with proper acknowledgement of their limits in capturing change. Having said that, I am still a little concerned about how change will be inferred in this review, as the inclusion criteria make no reference to studies having to be about change (just "impact", which is vaguer). Besides using the word "impact" the only relevant inclusion statement is "contribute directly to understanding gender disparities within households during the pandemic", which doesn't entail any reference to change during or as a result of the pandemic, real or perceived. Studies that merely contribute to understanding disparities in the pandemic's present (disparities that may have been pre-existing) will therefore meet inclusion criteria. So I remain a little uncertain how the findings will illuminate change unless the included studies at the very least examine perceived change from before to during (or after) the pandemic. That doesn't currently seem to be a strict requirement of included studies, so I remain a little doubtful how the review will clarify change or impact unless the inclusion criteria are tightened a little.

7. PLOS authors have the option to publish the peer review history of their article (what does this mean?). If published, this will include your full peer review and any attached files.

Reviewer #1: No

Reviewer #2: No

---

## [Author Response · Author response to Decision Letter 1]

16 Oct 2024

Journal: PLOS One

PONE-D-24-14476

 15-10-2024

Dear Editor,

On behalf of all co-authors, we would like to thank you and the reviewers for the time spent on reviewing our work, as well as the valuable comments that helped us improve the quality of our manuscript titled “Impact of the COVID-19 Pandemic on Intra-household Gender Disparities in the Middle East and North Africa region: A Scoping Review Protocol”. All comments have been addressed, as can be seen in the point-by-point response to the comments by reviewers below.

All authors confirm that the manuscript is not under consideration by any other journal and that authors have no conflict of interest. All authors have read and approved the submission of this revised manuscript.

Yours sincerely, 

Jocelyn DeJong, PhD

Associate Provost and Professor at the Faculty of Health Sciences, American University of Beirut, Beirut, Lebanon 

Email: Jd16@aub.edu.lb

Journal requirements

Thank you for the comment. The references have been revised. 

Reviewer #1

We would like to thank the Reviewer for the time reviewing our work and the valuable comments.

Reviewer #1: This is a review of a revised scoping study protocol. I think the authors have given careful consideration to the comments from myself and another reviewer, which have significantly improved the intended scoping review. I think this has the potential to be a very important study, providing a much needed contribution to the literature. I would be happy for the authors to proceed, with just two minor comments.

As noted by R2, the first mention of LMIC should be spelled out fully. This currently takes place on p4, but is not spelled out until p5.

Thank you for your comment. The manuscript has been edited accordingly (p. 4, line 106).

On p7, the authors provide a careful outline of the themes to be included in the review. However, no references are provided. These should be included to understand the basis for including these themes.

Thank you for your valuable feedback. We have incorporated relevant references on page 7 to support the themes outlined in the review. These citations provide a clear rationale for the selection of these themes, drawing on key studies and existing literature that highlight their importance (Lines 176-255). 

Reviewer #2

We would like to thank the Reviewer for the time spent on our work and the valuable comments.

Reviewer #2: The revised versions is largely responsive to my original comments. I'll discuss each in turn, numbered in the same way.

1. Dealt with appropriately.

Thank you for your comment.

2. This is a positive change and should increase the yield of useable studies.

Thank you for your comment.

3. These are much improved. I was only seeking clarification on what each theme was intended to cover, and the new text also provides a rationale for doing so, but I see the other reviewer requested this so I am satisfied with it.

Thank you for your comment.

4. I appreciate that there may be a paucity of longitudinal studies and that it is therefore reasonable to include cross-sectional studies with proper acknowledgement of their limits in capturing change. Having said that, I am still a little concerned about how change will be inferred in this review, as the inclusion criteria make no reference to studies having to be about change (just "impact", which is vaguer). Besides using the word "impact" the only relevant inclusion statement is "contribute directly to understanding gender disparities within households during the pandemic", which doesn't entail any reference to change during or as a result of the pandemic, real or perceived. Studies that merely contribute to understanding disparities in the pandemic's present (disparities that may have been pre-existing) will therefore meet inclusion criteria. So I remain a little uncertain how the findings will illuminate change unless the included studies at the very least examine perceived change from before to during (or after) the pandemic. That doesn't currently seem to be a strict requirement of included studies, so I remain a little doubtful how the review will clarify change or impact unless the inclusion criteria are tightened a little.

Thank you for your insightful feedback regarding our inclusion criteria and the inference of change in our review. In light of your comments, we have refined our research question and themes to ensure a clearer focus on both the characterization of the present data and the identification of studies that report change related to intra-household gender disparities during the COVID-19 pandemic. Our revised research question now explicitly asks: “What are the types, characteristics, and implications of intra-household gender disparities identified during the COVID-19 pandemic in LMICs in the MENA region?” (Lines 128-132).

---

## [Decision Letter · Decision Letter 2]

1 Nov 2024

Impact of the COVID-19 Pandemic on Intra-household Gender Disparities in the Middle East and North Africa region: A Scoping Review Protocol

PONE-D-24-14476R2

Dear Dr. DeJong,

We’re pleased to inform you that your manuscript has been judged scientifically suitable for publication and will be formally accepted for publication once it meets all outstanding technical requirements.

Kind regards,

Amin Nakhostin-Ansari

Academic Editor

PLOS ONE

Additional Editor Comments (optional):

Reviewers' comments:

Reviewer's Responses to Questions

**Comments to the Author**

1. Does the manuscript provide a valid rationale for the proposed study, with clearly identified and justified research questions?

Reviewer #1: Yes

Reviewer #2: Yes

2. Is the protocol technically sound and planned in a manner that will lead to a meaningful outcome and allow testing the stated hypotheses?

Reviewer #1: Yes

Reviewer #2: Yes

3. Is the methodology feasible and described in sufficient detail to allow the work to be replicable?

Reviewer #1: Yes

Reviewer #2: Yes

4. Have the authors described where all data underlying the findings will be made available when the study is complete?

Reviewer #1: Yes

Reviewer #2: Yes

5. Is the manuscript presented in an intelligible fashion and written in standard English?

Reviewer #1: Yes

Reviewer #2: Yes

6. Review Comments to the Author

You may also provide optional suggestions and comments to authors that they might find helpful in planning their study.

Reviewer #1: Thank you for addressing all of the reviewer comments. I am satisfied that all issues have been addressed and look forward to seeing the study.

Reviewer #2: The authors have responded reasonably to my final concern, essentially by backing away from the claim that they will be examining change. The new question -- “What are the types, characteristics, and implications of intra-household gender disparities identified during the COVID-19 pandemic in LMICs in the MENA region?” -- doesn't require that the disparities emerged during or in response to the pandemic. That's ok, but I do hope when writing up the research they are careful when making claims about whether or not they have really documented changes or just, perhaps, pre-existing tendencies.

7. PLOS authors have the option to publish the peer review history of their article (what does this mean?). If published, this will include your full peer review and any attached files.

Reviewer #1: No

Reviewer #2: No

---

## [Editor Report · Acceptance letter]

7 Nov 2024

PONE-D-24-14476R2 

PLOS ONE

Dear Dr. DeJong, 

I'm pleased to inform you that your manuscript has been deemed suitable for publication in PLOS ONE. Congratulations! Your manuscript is now being handed over to our production team.

Kind regards, 

on behalf of

Dr. Amin Nakhostin-Ansari 

Academic Editor

PLOS ONE